# Endometriosis Associated-miRNome Analysis of Blood Samples: A Prospective Study

**DOI:** 10.3390/diagnostics12051150

**Published:** 2022-05-05

**Authors:** Sofiane Bendifallah, Yohann Dabi, Stéphane Suisse, Léa Delbos, Mathieu Poilblanc, Philippe Descamps, Francois Golfier, Ludmila Jornea, Delphine Bouteiller, Cyril Touboul, Anne Puchar, Emile Daraï

**Affiliations:** 1Clinical Research Group (GRC) Paris 6: Centre Expert Endométriose (C3E), Department of Obstetrics and Reproductive Medicine, Hôpital Tenon, Sorbonne University (GRC6 C3E SU), 4 rue de la Chine, 75020 Paris, France; yohann.dabi@gmail.com (Y.D.); cyril.touboul@gmail.com (C.T.); emile.darai@aphp.fr (E.D.); 2Cancer Biology and Therapeutics INSERM UMR_S_938, Centre de Recherche Saint-Antoine (CRSA), 75020 Paris, France; 3Ziwig, 19 rue Reboud, 69003 Lyon, France; stephane@ziwig.com; 4Endometriosis Expert Center—Pays de la Loire, Department of Obstetrics and Reproductive Medicine—CHU d’Angers, 49100 Angers, France; lea.delbos@chu-angers.fr (L.D.); phdescamps@chu-angers.fr (P.D.); 5Endometriosis Expert Center—Steering Committee of the EndAURA Network, Department of Obstetrics and Reproductive Medicine, Lyon South University Hospital, Lyon Civil Hospices, 69310 Pierre Bénite, France; mathieupoilblanc@gmail.com (M.P.); francois.golfier@chu-lyon.fr (F.G.); 6Paris Brain Institute—Institut du Cerveau—ICM, Inserm U1127, CNRS UMR 7225, AP-HP—Hôpital Pitié-Salpêtrière, Sorbonne University, 75006 Paris, France; ludmila.jornea@icm-institute.org; 7Genotyping and Sequencing Core Facility, iGenSeq, Institut du Cerveau et de la Moelle épinière, ICM, Hôpital Pitié-Salpêtrière, 47-83 Boulevard de l’Hôpital, 75013 Paris, France; delphine.bouteiller@icm-institute.org; 8Department of Obstetrics and Reproductive Medicine, Hôpital Tenon, Sorbonne University, 4 rue de la Chine, 75020 Paris, France; anne.puchar@aphp.fr

**Keywords:** endometriosis, miRNA, NGS, bioinformatics

## Abstract

The aim of our study was to describe the bioinformatics approach to analyze miRNome with Next Generation Sequencing (NGS) of 200 plasma samples from patients with and without endometriosis. Patients were prospectively included in the ENDO-miRNA study that selected patients with pelvic pain suggestive of endometriosis. miRNA sequencing was performed using an Novaseq6000 sequencer (Illumina, San Diego, CA, USA). Small RNA-seq of 200 plasma samples yielded ~4228 M raw sequencing reads. A total of 2633 miRNAs were found differentially expressed. Among them, 8.6% (*n* = 229) were up- or downregulated. For these 229 miRNAs, the F1-score, sensitivity, specificity, and AUC ranged from 0–88.2%, 0–99.4%, 4.3–100%, and 41.5–68%, respectively. Utilizing the combined bioinformatic and NGS approach, a specific and broad panel of miRNAs was detected as being potentially suitable for building a blood signature of endometriosis.

## 1. Introduction

The mammalian genome contains sequences for RNAs coding for messenger RNA (mRNA) proteins and non-coding RNAs (ncRNA). ncRNAs represent 98% of the transcriptome [1]. The known ncRNAs are subdivided into long non-coding RNAs (lncRNAs comprising more than 200 nucleotides (nt)) and small non-coding RNAs (sncRNAs) comprising less than 50 nucleotides [2,3]. To date, miRNAs have been studied far more than their non-coding counterparts. However, during the last decades, a growing interest appears for ncRNAs because of their implication in many benign, malignant pathologies and also in neurodegenerative diseases.

Approximately 70% of studies evaluated exosomes as the source of choice for ncRNAs [4]. The RNA content in the exosomes is estimated at 40.4% mature miRNAs, 40% piwi-interacting RNAs (piRNAs), 3.7% pseudogenes, 2.4% lncRNAs, tRNAs at 2.1%, and mRNAs at 2.1% of total RNA [5]. Among sncRNAs, numerous studies have focused on the role of miRNAs, which are detectable in various body fluids, as potential biomarkers for various pathologies [6,7,8]. Currently, more than 2600 human miRNAs have been annotated [9,10,11,12]. miRNAs are single-stranded conserved sncRNAs composed of 21–25 nucleotides playing a pivotal role in gene degradation and translation by binding to their complementary messenger RNA (mRNA) [13]. The human miRNA spectrum varies according to cell type, tissue type, developmental stage, environmental factors, and health/disease state and disease stages [14,15,16,17]. Recently, numerous studies have demonstrated the values of miRNAs in various cancers and systemic disease, but rarely in the context of endometriosis [8,13,16,17].

Endometriosis, defined by the presence of endometrium-like tissue outside the uterus, affects 2–10% of the female population, i.e., around 190 million women worldwide [18]. It is well known that endometriosis is a debilitating disease associated with severe symptoms. Consequently, endometriosis negatively affects all aspects of quality of life and is considered a public health issue related to its socioeconomic impact, and treatment and clinical management costs [19,20,21]. Previous studies have evaluated the potential of circulating miRNAs as biomarkers for endometriosis [22] and association with functions and pathophysiological pathways in endometriosis [8,15,17,23] but with conflicting results. This is due firstly to pre-analytical factors such as the source of the miRNAs (serum or plasma), type of blood collection tubes (EDTA versus heparin), hemolysis, and sample processing protocols. Secondly, there are technical factors related to the method used for RNA extraction, miRNA expression analysis (microarray, qRT-PCR and next-generation sequencing (NGS) techniques), and the strategy for normalization of miRNA expression data. Finally, biological factors are also implicated: the genetic background of the study cohort, the control population (self-reported healthy versus laparoscopically proven absence of endometriosis), and the extent of endometriosis (stage I versus stage IV) [8,15,17].

Therefore, the goal of the present study was to describe using NGS and bioinformatics systematic approach the miRNome sequencing of 200 plasma samples based on the prospective data from the ENDO-miRNA study.

## 2. Material and Methods

### 2.1. Study Population

We used data from the prospective “ENDO-miRNA” study (ClinicalTrials.gov Identifier: NCT04728152) [24]. Data collection and analysis (previously presented) were carried out under Research Protocol n° ID RCB: 2020-A03297-32 [25]. The IRB was delivered by the Comité de Protection des Personnes (C.P.P.) Sud-Ouest et Outre-Mer 1 (CPP 1-20-095 ID 10476. All patients gave informed written consent. The ENDO-miRNA study included 200 plasma samples obtained from patients with chronic pelvic pain suggestive of endometriosis. All had undergone a laparoscopic procedure (either operative or diagnostic) and/or magnetic resonance imaging (MRI) imaging proving endometriosis by the presence of endometrioma and/or deep endometriosis [26,27,28]. All laparoscopies were performed by two expert surgeons in endometriosis (ED, SB). For these patients, diagnosis of endometriosis was confirmed by histology. For the patients without laparoscopic evaluation, endometriosis was diagnosed when MRI revealed features of deep endometriosis with colorectal involvement and/or endometrioma confirmed by a multidisciplinary endometriosis committee. The study population was eventually composed of two groups: (i) endometriosis group composed of patients with endometriosis confirmed at either laparoscopy or MRI; (ii) control group without endometriosis at laparoscopy with and without other gynecological disorders. All patients included in the control group underwent a systematic laparoscopy. Among patients of the endometriosis group, 83 (54.2%) underwent an operative laparoscopy with histological confirmation of endometriosis and the remaining 70 (45.8%) had MRI confirmation [29]. The samples were collected from all the participants between January 2021 and June 2021. Statistical and miRNAs assays analysis were performed blinded to the surgical and imaging findings. The patients with endometriosis were stratified according to the revised American Society of Reproductive Medicine (rASRM) classification [30]. All patients filled online questionnaires to assess their symptoms and intensity using Visual Analogic Scale (VAS) [31].

### 2.2. Sample Collection

Blood samples (4 mL) were collected in EDTA tubes (BD, Franklin Lakes, NJ, USA). Plasma was isolated from whole blood within 2 h after blood sampling by two successive centrifugations at 4 °C (first at 1900× *g* (3000 rpm) for 10 min, followed by 13,000–14,000× *g* for 10 min to remove all cell debris) then aliquoted, labeled, and stored at −80 °C until analysis, as previously published [32,33,34].

### 2.3. RNA Sample Extraction, Preparation and Quality Control

RNA was extracted from 500 μL of plasma on a Maxwell 48^®^ RSC automat using the Maxwell^®^ RSC miRNA Plasma and Serum Kit (ref AS1680, Promega, USA) according to the manufacturer’s protocol. Libraries for small RNA sequencing were prepared using the QIAseq miRNA Library Kit for Illumina (Qiagen, Hilden, Germany). The resulting small RNA libraries were concentrated by ethanol precipitation and quantified using a Qubit 2.0 Fluorometer (Thermo Fisher Scientific, Waltham, MA, USA). Samples were indexed in batches of 96, with a targeted sequencing depth of 17 million reads per sample. Sequencing was performed using 100 base single-end reads, using an Novaseq6000 sequencer (Illumina, San Diego, CA, USA) [35,36].

## 3. Bioinformatics 

### 3.1. Raw Data Preprocessing (Raw, Filtered, Aligned Reads) and Quality Control

Sequencing reads were processed after adaptation using the bioinformatics and processing pipeline according to Potla et al., review [37,38]. FastQ files were trimmed to remove adapter sequences using Cutadapt version v.1.18 and were aligned using Bowtie version 1.1.1 to the following transcriptome databases: the human reference genome available from NCBI (https://www.ncbi.nlm.nih.gov/genome/guide/human/ 10 March 2022 and miRBase21) (miRNAs) using the MirDeep2 v0.1.0 package. The raw sequencing data quality was assessed using FastQC software v0.11.7 [10,14,35,39,40]. 

### 3.2. Differential Expression Analysis of miRNA

miRNA expression was quantified using miRDeep2 [41]. Differential expression tests were then conducted in DESeq2 for miRNAs with read counts in ≥1 of the samples. DESeq2 integrates methodological advances with several novel features to facilitate a more quantitative analysis of comparative RNA-seq data using shrinkage estimators for dispersion and fold change [41,42]. miRNAs were considered as differentially expressed if the absolute value of log2-fold change was >1.5 (up) and <0.5 (down) and the *p* value adjusted for multiple testing was <0.05 [41].

### 3.3. miRNome Accuracy

To evaluate the accuracy of each miRNA biomarker, sensitivity, specificity, and ROC analysis was performed, and the ROC AUC was calculated [43,44]. Additional statistical analysis was based on the Chi^2^ test as appropriate for categorical variables. Values of *p* < 0.05 were considered to denote significant differences. Data were managed with an Excel database (Microsoft, Redmond, WA, USA) and analyzed using R 2.15 software, available online (http://cran.r-project.org/, accessed on 10 March 2022).

## 4. Results

### 4.1. Description of the ENDO-miRNA Cohort

The clinical characteristics of the endometriosis and control patients are presented in Table 1. Among the 200 patients, 76.5% (*n* = 153) were diagnosed with endometriosis and 23.5% (*n* = 47) without. In the endometriosis group, 52% (80) had rASRM stages I–II and 48% (73) had stages III–IV. The control group was mainly composed of complex patients defined by patients sharing symptoms of endometriosis but without clinical or imaging features of endometriosis, and patients with other gynecologic disorders but with symptoms suggestive of endometriosis.

### 4.2. Global Overview of miRNA Transcriptome

Small RNA-seq of 200 plasma samples yielded ~4228 M raw sequencing reads (from ~11.7 M to ~34.98 M reads/sample). Pre-filtering and filtering steps retained 39% (~1639 M) of initial raw reads. The majority of filtered reads were of 20–23 nt length which corresponds to the range of mature miRNA sequences. Quantification of filtered reads and identification of known miRNAs yielded ~2588 M sequences to be mapped to 2633 known miRNAs from miRBase v22. The number of expressed miRNAs ranged from 666 to 1274 per sample. The distribution of expressed miRNAs in the 200 plasma samples and according to the overall composition of processed reads is shown in Figure 1A,B and Figure 2.

### 4.3. miRNA Expression in Patients with and without Endometriosis

A total of 2633 miRNAs were found differentially expressed in the plasma samples of patients with endometriosis compared with control patients. Among these, 8.6% (*n* = 229) were up- or downregulated. Respectively, 66% (152) and 34% (77) of the 229 miRNAs were up- and down regulated (Appendix A). A volcano plot of the expressed miRNAs in the endometriosis patients is reported in Figure 3. Among the 152 miRNAs upregulated, only 5 (hsa-miR-29b-1-5p, hsa-miR-4748, hsa-miR-515-5p, hsa-miR-548j-5p, hsa-miR-6502-5p) had an AUC > 0.6. Among the 77 miRNAs downregulated, 2 (hsa-miR-3137, hsa-miR-3168) had an AUC > 0.6.

### 4.4. Diagnostic Accuracy of Regulated miRNAs

The diagnostic metrics for endometriosis for all the regulated plasma miRNAs (*n* = 229) are reported in Appendix A. Among these 229 miRNAs, the F1-score, sensitivity, specificity, and AUC ranged from 0–88.2%, 0–99.4%, 4.3–100%, and 41.5–68%, respectively.

For AUC criteria, 96.9% (222) and 3.1% (7) had a value ranging between 41.5–59% and ≥60%, respectively.

For the F1-scores, 66.8% (*n* = 153) and 33.2% (*n* = 76) had a value ranging between 0–79%, and ≥80%, respectively.

For sensitivity, 69.8% (*n* = 160) and 30.1% (*n* = 69) had a value ranging between 0–79%, and ≥80%, respectively.

For specificity, 42% (*n* = 96) and 58% (*n* = 133) had a value ranging between 0–79%, and ≥80%, respectively.

Among the 229 regulated miRNAs, 69 had a sensitivity over 0.80 and 100 had a specificity over 0.80 but none had both sensitivity and specificity over 0.80 (Appendix A).

## 5. Discussion

To our knowledge this is the first report which describes a sequencing and systematic bioinformatics approach for plasma miRNome of patients with endometriosis. The current study demonstrates that using an NGS technique allows the display of a specific and broad panel of miRNAs potentially suitable for statistical analysis as potential biomarker in endometriosis.

The recent literature review by Monnaka et al. of miRNA expression in endometriosis found that 30 miRNAs were deregulated in the blood; 27 in the serum, and 18 in the plasma of women with endometriosis compared with control populations. Interestingly, the authors of this critical review concluded that no particular miRNAs or miRNA combination was individually accurate enough to screen and diagnose endometriosis [17]. Similarly, Vanhie et al., reported that 42 miRNAs were deregulated in the blood samples of a biobank of patients with endometriosis, but failed to build a signature [15]. In addition, several miRNAs have been shown to be deregulated during the pathogenic process of endometriosis [8,45,46]. For example, Maged et al. have shown that serum miR-122 and miR-199a had a sensitivity of 95.6 and 100.0% and a specificity of 91.4 and 100%, respectively, for diagnosis of disease status in women [47]. All these data raise the question of the technical and methodological obstacles to identify miRNAs significantly associated with the presence of endometriosis. Among these obstacles, the technology used to investigate circulatory miRNAs appears crucial. Indeed, most previous studies evaluated miRNA profiles by microarray. Subsequently, only the most differentially expressed miRNAs in patients with endometriosis were validated by qRT-PCR which represents a major bias [8]. Moreover, fold change varies from one series to another, or is not mentioned, and AUC was not systematically reported [48,49]. In addition, few numbers of miRNA biomarkers have been studied in contrast with the large number of miRNAs associated with endometriosis assessed in the current study [17,22]. To overcome these concerns, in the current prospective study miRNAs were sequenced using NGS platforms, allowing the analysis of millions of RNA fragments. Moreover, unlike microarray, the NGS technique—recognized as one of the most efficient tools in this domain—does not require sequence specific hybridization probes [8]. Thanks to this technology, the 200 plasma samples we analyzed initially yielded ~4228 M raw sequencing reads (from ~11.7 M to ~34.98 M reads/sample) of which 39% (~1639 M) were retained after the pre-filtering and filtering steps. Among the 2633 miRNAs expressed in the plasma of patients with endometriosis compared to control patients, 229 were up- or downregulated. Of these, 30.1% had a sensitivity ≥80%, and 58% had a specificity ≥80%. The rate of miRNAs with an AUC value ≥ 60% was 3.1%. These values attest that high quality and high yields of transcriptomic miRNA information can be isolated from plasma without the need for preamplification. Another crucial goal is to evaluate the stability and the reproducibility of the miRNA reads for the 200 samples (100% of sample). In the current study, these two criteria were fulfilled: all 200 samples were used for sequencing, and bioinformatics treatment provided diagnostic accuracies according to the F1-score, sensitivity, specificity, and AUC, which ranged from 0–88.2%, 0–99.4%, 4.3–100%, and 41.5–68%, respectively.

Another issue is the difficulty of simultaneously analyzing hundreds of miRNAs taking into account the diversity of endometriosis phenotypes and the incomplete knowledge of the pathophysiology [22]. Previous studies have focused on the miRNAs involved in classic known signaling pathways of endometriosis including proliferation, apoptosis, cell differentiation, angiogenesis, inflammation, etc. [8,17,50]. While this approach is logical for a disorder with well identified pathophysiologic mechanisms, it is not suitable in the specific setting of endometriosis with incomplete knowledge of signaling pathways. In this specific issue, additional research are required to evaluate the relationship between miRNAs expression and genetic, epigenetic, and metabolomic abnormalities [51,52,53]. This suggests that there are issues concerning the methodology used to select the miRNAs, as well as the characteristics of the control group. As previously shown for cancer, to build a miRNA signature sufficiently stable to provide the same accuracy across different platforms implies the inclusion of a broad spectrum of miRNAs [14,54]. This is totally in keeping with previous studies [14,54,55,56]: it is illusory to attempt to reflect the heterogeneity of a multifactorial disorder such as endometriosis by a limited number of miRNAs. Therefore, beyond classis logistic regression, it is necessary to use both NGS and new mathematical tools such as Machine Learning (ML) and Artificial Intelligence (AI) as proved in cancer models [14,54,55,57,58].

Another issue is the discrepancy in miRNA expression from one series to another [8,15,17,23]. In addition to endogenous qRT-PCR controls, and the platforms used for miRNA analysis, these discrepancies could be related to various causes such as differences in study design, patient population, sample size, and stage of endometriosis, but mainly to the composition of the control group. As underlined by Agrawal et al., choosing an appropriate control group is challenging and crucial to study miRNA expression. To limit the risk of bias, patients with pelvic inflammatory disease or autoimmune disorders, as well as healthy women that may have asymptomatic endometriosis (unless ruled out by laparoscopic evaluation) should be excluded [59]. We specifically designed a prospective study, including 200 patients—the largest series to date—able to quantify the miRNome for (i) complex patients (women with pelvic pain suggestive of endometriosis and both negative clinical and imaging findings with and without endometriosis at diagnostic laparoscopy), (ii) women with early-stage endometriosis (stage I-II rASRM), (iii) women with advanced stages (stage III-IV rASRM), (iv) women with various phenotypes of endometriosis (peritoneal endometriosis, ovarian endometriosis also called endometrioma, deep endometriosis defined by infiltration of pelvic organs and/or pelvic anatomical structures with and without endometrioma), and (v) women with other gynecologic disorders sharing symptoms of endometriosis.

Some remaining issues for the evaluation of miRNAs in the context of endometriosis should be discussed. Two of the important determinants of miRNA analysis are the phase of the menstrual cycle and the impact of hormonal treatments [60]. Although several studies have observed differences in miRNA expression in tissues according to the menstrual phase mainly at endometrial level [8,61,62,63], no such cyclic differences were observed in the plasma of healthy women [49]. One hypothesis is that changes in miRNA expression at the endometrium level regulate gene expression locally but are insufficient to cause detectable systemic changes [8]. Moreover, previous studies report no specific impact of hormonal treatment on blood miRNA [8,15]. Another concern is that among the sncRNAs, we only focused on miRNA while it is possible that other RNAs such as siRNAs, piRNAs, and snoRNAs as well as lncRNAs could be of diagnostic and therapeutic values.

## 6. Future Perspectives

It is well known that endometriosis, which affects 190 million women worldwide, is a debilitating disorder with a negative impact on quality of life and fertility, and that it represents a major socioeconomic burden [64]. Our results provide further evidence that patients with endometriosis exhibit a specific panel of miRNAs potentially suitable as biomarker. Beyond the specific context of endometriosis, the methodology developed in the current study can be transposable to other benign chronic and malignant diseases.

## Figures and Tables

**Figure 1 diagnostics-12-01150-f001:**
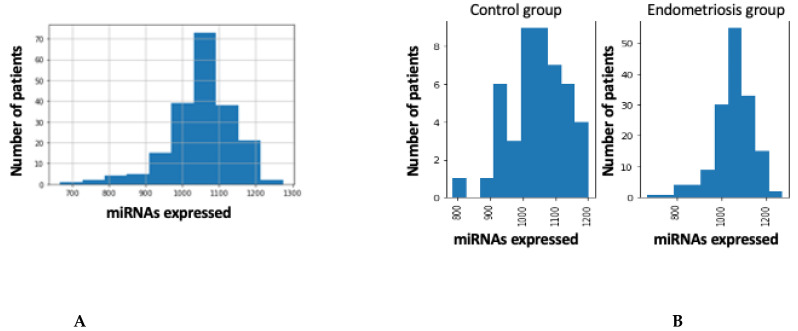
(**A**) Distribution of expressed miRNAs in the 200 blood samples. (**B**) Distribution of expressed miRNAs in the samples diagnosis.

**Figure 2 diagnostics-12-01150-f002:**
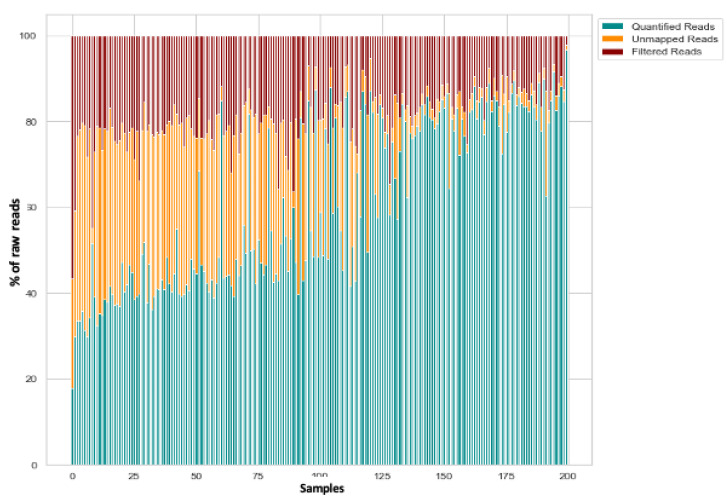
Overall composition of processed reads for plasma samples. In green: quantified reads; In Yellow: Unmapped reads; In Red: Filtered reads.

**Figure 3 diagnostics-12-01150-f003:**
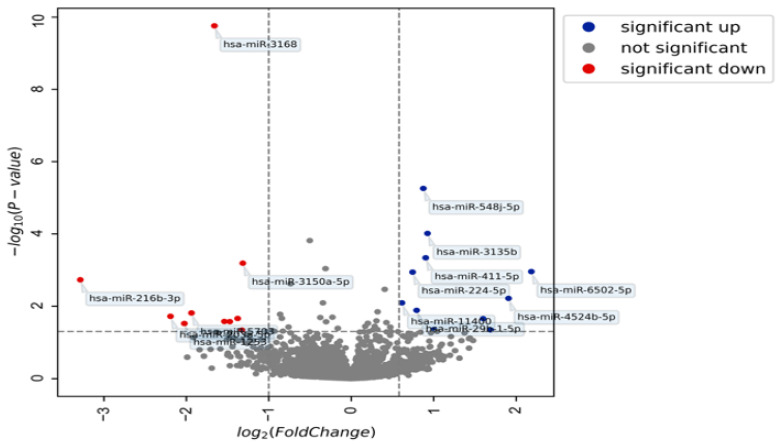
Volcano plot of expressed miRNAs in plasma for endometriosis.

**Table 1 diagnostics-12-01150-t001:** Main characteristics of the patients included.

	Controls N = 47	Endometriosis N = 153	*p* Value
**Age years (mean ± SD)**	30.92 ± 13.79	31.17 ± 10.78	0.19
**BMI (body mass index) (mean ± SD)**	24.84 ± 11.10	24.36 ± 8.38	0.52
**rASRM classification**			
- I–II	-	80 (52%)	
- III–IV	-	73 (48%)	
**Control diagnoses**			
- No abnormality	24 (51%)	-	-
- Leiomyoma	1 (2%)		
- Cystadenoma	5 (11%)		
- Teratoma	11 (23%)		
- Others gynecological disorders	6 (13%)		
**Dysmenorrhea**	100%	100%	
**Abdominal pain outside menstruation**			
- Yes	21 (66%)	89 (71%)	0.69
**Patients with pain suggesting sciatica**	10 (31%)	70 (56%)	**0.02**
**Dyspareunia intensity at VAS (mean ± SD)**	4.95 ± 3.52	5.28 ± 3.95	**<0.001**
**Patients with lower back pain outside menstruation**	20 (62%)	101 (81%)	**0.049**
**Intensity of pain during defecation at VAS (mean ± SD)**	2.84 ± 2.76	4.35 ± 3.47	**<0.001**
**Patient with right shoulder pain during menstruation**	3 (9%)	26 (21%)	0.21
**Intensity of urinary pain during menstruation at VAS (mean ± SD)**	2.84 ± 2.76	4.35 ± 3.36	**<0.001**
**Patient with blood in the stools during menstruation**	4 (12%)	30 (24%)	0.24
**Patient with blood in urine during menstruation**	8 (25%)	21 (17%)	0.41

## Data Availability

All relevant data are within the manuscript or Appendix A.

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
