# Peer review of "Endometriosis Associated-miRNome Analysis of Blood Samples: A Prospective Study"

_diagnostics, 2022, doi:10.3390/diagnostics12051150_

Round 1

Reviewer 1 Report

The manuscript on the use of bioinformatics and miRNA signatures among patients with and without endometriosis was very interesting to read, especially from the perspective of expanding on miRNA analyses within endometriosis research, which has been understudied compared to other conditions. While I feel the content of the paper will be useful for the readership of Diagnostics, there are some improvements that could be made to the manuscript.

Abstract

  1. It would be clearer if the authors changed ‘Thank to the combined…’ on Line 63 to ‘Utilizing the combined…’

Introduction

  1. The introduction was long and contained information that was not essential to the manuscript. I would suggest removing a lot of the background information on the different types of RNAs and focus mainly on miRNAs.
  2. Page 4 Line 95 – By censured, do you mean identified and cataloged?
  3. Page 5 Lines 102-103 – Please provide a reference for this sentence.

Methods

  1. Page 6 Line 126 – Please include a reference for the previous paper that has more details about data collection and analysis.
  2. Page 6 Lines 126-127 – Please include the institution that approved the IRB protocol. Did the participants provide written informed consent?
  3. Page 6 Line 132 – It would be useful for the reader to add into the text if any of the endometriosis patients who had endometriosis visualized at surgery but not histologically confirmed. Additionally, it would be useful to know how many participants had surgery and how many had MRIs for both those with endometriosis and those without.
  4. Page 6 Lines 136-137 – Were the statistical analyses blinded to surgical and imaging findings or were the miRNA assays blinded to surgical and imaging findings, or both? Please clarify in the text.
  5. Within the methods section, it would be helpful to explicitly state that participants found to not have endometriosis were considered controls within the study.
  6. Please include information within the methods section on how pain symptoms information shown in Table 1 was collected from participants.

Results

  1. Figures 1a and b – These figures need to have y and x axis titles added to them. It is also not clear what is meant by ‘0’ and ‘1’ at the top of the two graphs for Figure 1b, please replace with the appropriate labels.
  2. Page 12 – The summary of these results may be easier to interpret in table format. And it would be interesting to know if there were any miRNAs that had both a high sensitivity and specificity.

Discussion

  1. Page 15 Lines 295-298 – The authors raise good points here about the potential bias that could be encountered depending on the control group used for miRNA studies (and all studies of endometriosis). One further limitation is not mentioned in the manuscript. When including a control group of patients with endometriosis symptoms who had surgery to rule out the presence of endometriosis, the results for such studies will provide information on miRNA differences between patients with endometriosis and those with other pelvic pathologies/symptoms. While this information may be useful for distinguishing those with symptoms severe enough to go to surgery who may have endometriosis, it will not necessarily provide useful information on endometriosis pathophysiology as similar miRNA dysregulation may occur for multiple pelvic pathologies.
  2. Page 16 Line 317 – I suggest replacing the word theragnostic with therapy as diagnostic is already included in the sentence.

Throughout manuscript

  1. There are abbreviations in the manuscript (e.g. MRI, nt, VAS) that have not been defined before the first time they are used in the paper.

Author Response

Reviewer 1

The manuscript on the use of bioinformatics and miRNA signatures among patients with and without endometriosis was very interesting to read, especially from the perspective of expanding on miRNA analyses within endometriosis research, which has been understudied compared to other conditions. While I feel the content of the paper will be useful for the readership of Diagnostics, there are some improvements that could be made to the manuscript.

Thank you for this positive comment.

Abstract

  1. It would be clearer if the authors changed ‘Thank to the combined…’ on Line 63 to ‘Utilizing the combined…’

As suggested, we modified the term used in the abstract.

Introduction

  1. The introduction was long and contained information that was not essential to the manuscript. I would suggest removing a lot of the background information on the different types of RNAs and focus mainly on miRNAs.

As suggested, the following section was suppressed from the revised version of the manuscript:

Among the sncRNAs, circular RNAs (circRNAs); generated by retro-splicing pre-mRNAs, small non-coding microRNAs (miRNAs), Piwi-interacting RNAs (piRNAs), transfer RNAs (tRNAs), small nuclear RNAs (snRNAs) and small interfering RNA (siRNA) have been identified [1]. A recent study reported the presence of tRNA-derived partial sncRNAs, such as half tRNA (tiRNA) and tRNA fragments (tRF) [2] and natural antisense transcripts (NAT; generated by transcription in the opposite direction to the protein-coding transcripts). Among the lncRNAs, a distinction is made between intergenic RNAs (lincRNA), circular RNAs (circRNA) and ribosomal RNAs (rRNA) [3].

  1. Page 4 Line 95 – By censured, do you mean identified and cataloged?

Indeed, that’s what we meant. We rephrased using the word “annotated” to clarify this part.

  1. Page 5 Lines 102-103 – Please provide a reference for this sentence.

A valuable reference was added :

  1. Zondervan, K.T.; Becker, C.M.; Missmer, S.A. Endometriosis. N Engl J Med 2020, 382, 1244–1256, doi:10.1056/NEJMra1810764.

Methods

  1. Page 6 Line 126 – Please include a reference for the previous paper that has more details about data collection and analysis.

As suggested, we added the references that detailed the protocol as well as data collection and previous analysis.

We used data from the prospective “ENDO-miRNA” study (ClinicalTrials.gov Identifier: NCT04728152) [27]. Data collection and analysis (previously presented) were carried out under Research Protocol n° ID RCB: 2020-A03297-32 [28].

References:

  1. Bendifallah, S. Evaluation of MiRNAs in Endometriosis; clinicaltrials.gov, 2021;
  2. Bendifallah, S.; Dabi, Y.; Suisse, S.; Jornea, L.; Bouteiller, D.; Touboul, C.; Puchar, A.; Daraï, E. MicroRNome Analysis Generates a Blood-Based Signature for Endometriosis. Sci Rep 2022, 12, 4051, doi:10.1038/s41598-022-07771-7.

  1. Page 6 Lines 126-127 – Please include the institution that approved the IRB protocol. Did the participants provide written informed consent?

The Institution that delivered the IRB was : “Comité de Protection des Personnes (C.P.P.) Sud-Ouest et Outre-Mer 1” (CPP 1-20-095 ID 10476)

Moreover, we clarified that all patients included gave written consent to be involved, in accordance with the CPP.

  1. Page 6 Line 132 – It would be useful for the reader to add into the text if any of the endometriosis patients who had endometriosis visualized at surgery but not histologically confirmed. Additionally, it would be useful to know how many participants had surgery and how many had MRIs for both those with endometriosis and those without.

All patients with the “endometriosis group” had histologically confirmed endometriosis.

All patients included in the control group underwent a systematic laparoscopy. Among patients of the endometriosis group, 83 (54.2%) underwent an operative laparoscopy with histological confirmation of endometriosis and the remaining 70 (45.8%) had MRI confirmation.

These data has been added in the revised version along with the reference of the initial publication that described the cohort study.

  1. Page 6 Lines 136-137 – Were the statistical analyses blinded to surgical and imaging findings or were the miRNA assays blinded to surgical and imaging findings, or both? Please clarify in the text.

Indeed, both analysis (statistical and miRNA assays) were performed completely blinded of the diagnosis

This was clarified in the revised version of the manuscript:

“Statistical and miRNAs assays analysis were performed blinded to the surgical and imaging findings.”

  1. Within the methods section, it would be helpful to explicitly state that participants found to not have endometriosis were considered controls within the study.

We clarified this part in the revised version of the manuscript.

The study population was eventually composed of two groups: i) endometriosis group composed of patients with endometriosis confirmed at either laparoscopy or MRI ii) control group without endometriosis at laparoscopy with and without other gynecological disorders.

  1. Please include information within the methods section on how pain symptoms information shown in Table 1 was collected from participants.

All patients included filled online questionnaires to assess their symptoms and intensity according to the IRB authorization delivered.

We added the reference in the manuscript.

Results

  1. Figures 1a and b – These figures need to have y and x axis titles added to them. It is also not clear what is meant by ‘0’ and ‘1’ at the top of the two graphs for Figure 1b, please replace with the appropriate labels.

As suggested we modified the figures

  1. Page 12 – The summary of these results may be easier to interpret in table format. And it would be interesting to know if there were any miRNAs that had both a high sensitivity and specificity.

In this study, among the 229 regulated miRNAs, 69 had a sensitivity over 0.80 and 100 had a specificity over 0.80 but none had both sensitivity and specificity over 0.80.

This sentence has been added in the revised version of the manuscript.

All sensitivity and specificity of the regulated miRNAs are displayed in supplementary table 1.

Discussion

  1. Page 15 Lines 295-298 – The authors raise good points here about the potential bias that could be encountered depending on the control group used for miRNA studies (and all studies of endometriosis).

One further limitation is not mentioned in the manuscript. When including a control group of patients with endometriosis symptoms who had surgery to rule out the presence of endometriosis, the results for such studies will provide information on miRNA differences between patients with endometriosis and those with other pelvic pathologies/symptoms. While this information may be useful for distinguishing those with symptoms severe enough to go to surgery who may have endometriosis, it will not necessarily provide useful information on endometriosis pathophysiology as similar miRNA dysregulation may occur for multiple pelvic pathologies.

Sorry, it is not clear for us whether the reviewer suggest a differential expression in miRNAs among the control group within patients with and without other benign gynecological conditions. For this specific question, no attempt was made to differentiate these two subgroups. However, previous studies have reported differential expression of miRNAs in patients with benign gynecological conditions such as mature teratoma or leiomyoma.

Effectively, some benign gynecological disorders may share the same miRNAs as these disorders might share the same physio-pathological pathways.

  1. Page 16 Line 317 – I suggest replacing the word theragnostic with therapy as diagnostic is already included in the sentence.

As suggested we modified the wording of the sentence.

Throughout manuscript

  1. There are abbreviations in the manuscript (e.g. MRI, nt, VAS) that have not been defined before the first time they are used in the paper.

We clarified the significance of the abbreviations used.

Reviewer 2 Report

I read with great interest the manuscript, which falls within the aim of this Journal. In my honest opinion, the topic is interesting enough to attract the readers’ attention. Authors should consider the following recommendations:

  • Manuscript should be further revised in order to correct some typos and improve style.
  • It would be appreciable if you can discuss the role of miRNA comparing it with those of  the genetic, epigenetic ed metabolomic theories (referring to PMID: 32143537; PMID: 30909768; PMID: 33174185)
  • the figures are not very clear and being so small they lose their optimal resolution

Author Response

I read with great interest the manuscript, which falls within the aim of this Journal. In my honest opinion, the topic is interesting enough to attract the readers’ attention.

Thank for this positive comment.

Authors should consider the following recommendations:

  • Manuscript should be further revised in order to correct some typos and improve style.

We performed a revision of the manuscript by an English – native speaker to improve the style. All typos were corrected

  • It would be appreciable if you can discuss the role of miRNA comparing it with those of the genetic, epigenetic ed metabolomic theories (referring to PMID: 32143537; PMID: 30909768; PMID: 33174185)

It is difficult in one sentence to Evoque the role of genetic, epigenetic and metabolomic theories suggested in the physiopathology of endometriosis. However, as suggested, we added a sentence to highlight the need for further research on this question with the valuable references provided by the reviewer.

In this specific issue, additional research are required to evaluate the relationship between miRNAs expression and genetic, epigenetic and metabolomic abnormalities [59–61].

  • the figures are not very clear and being so small they lose their optimal resolution

As suggested by the reviewer 1 also, we modified the figures and enlarged them.

Round 2

Reviewer 1 Report

The authors did a thorough job of revising the manuscript. I only have one minor point that it seems Figures 1a and 1b have not been updated in the revised manuscript to include the axis titles for the x- and y-axes.

Author Response

All authors would like to sincerely thank the reviewer. 

The revised figures have been incorporated in the manuscript.